# Consumption of β-Caryophyllene Increases the Mating Success of *Bactrocera zonata* Males (Diptera: Tephritidae)

**DOI:** 10.3390/insects15050310

**Published:** 2024-04-26

**Authors:** Ihsan ul Haq, Sehar Fatima, Awais Rasool, Todd E. Shelly

**Affiliations:** 1Insect Pest Management Program, National Agricultural Research Centre, Islamabad 45500, Pakistan; seharfatimauaf@gmail.com (S.F.); awaisrasool@hotmail.com (A.R.); 2USDA-APHIS, Waimanalo, HI 96795, USA; todd.e.shelly@usda.gov

**Keywords:** *Bactrocera zonata*, β-caryophyllene, mating success, sterile insect technique

## Abstract

**Simple Summary:**

The genus *Bactrocera* contains invasive fruit fly species that lay eggs in and consume many important fruits and vegetables. Two tactics used to control these flies include the male annihilation and the sterile insect techniques. *Bactrocera* males are attracted to and feed on methyl eugenol, a chemical found in many plants. Baiting traps with a mixture of this chemical and an insecticide are used to annihilate males and, thereby, the pest population. The release of sterile males (achieved via irradiation) results in mating with wild females that results in inviable embryos. Feeding on ME (without insecticide) enhances the mating success of *Bactrocera* spp. males and suppresses their subsequent attraction to methyl eugenol + insecticide traps, enabling the simultaneous application of the male annihilation and the sterile insect techniques, which will boost the efficiency of control programs. However, methyl eugenol may pose human health risks, and this study demonstrated that feeding on an alternative and safer plant compound, β-caryophyllene, similarly led to the increased mating success of *Bactrocera zonata* and reduced their subsequent attraction to methyl-eugenol-baited traps.

**Abstract:**

The peach fruit fly, *Bactrocera zonata* (Saunders) (Diptera: Tephritidae), is an economically important polyphagous quarantine pest of horticultural crops endemic to South and Southeast Asia. Methyl eugenol (ME), a naturally occurring phenylpropanoid, is a male attractant used to lure and (when mixed with an insecticide) annihilate the males from the wild population, a method of pest control termed the male annihilation technique (MAT). ME is reported to enhance the mating success of sterile males of *Bactrocera* spp., which is critical for enhancing the effectiveness of the sterile insect technique (SIT). The suppressed response of ME-treated males to ME-baited traps/devices allows the simultaneous application of the MAT and SIT, increasing the efficiency of area-wide integrated pest management (AW-IPM) programs. However, ME treatment in sterile males in SIT facilities is logistically difficult. β-caryophyllene (BCP) is a widely occurring, safer plant compound and is considered suitable for treating males in SIT facilities. Here, we demonstrate that BCP feeding enhanced *B. zonata* male mating success to the same extent as ME feeding. Feeding on BCP suppressed the male’s subsequent attraction to ME-baited traps, but not to the same degree as feeding on ME. The results are discussed and BCP is suggested as an alternative to ME for the concurrent use of the MAT and SIT.

## 1. Introduction

Plant-borne chemicals are known to influence the behavior of true fruit flies (Diptera: Tephritidae). For example, in addition to visual cues, gravid females utilize plant odors to locate suitable host plants for oviposition [1,2,3,4]. Field observations have also shown that males of certain tephritid species are attracted to and feed on the specific volatile compounds of certain plants, which may then—in modified or unmodified form—be sequestered and incorporated into the sex pheromone [5,6,7,8,9]. The best-known example involves males of the oriental fruit fly, *Bactrocera dorsalis* (Hendel), which are strongly attracted to methyl eugenol (ME), a phenylpropanoid that occurs in over 450 plant species [10]. Studies have demonstrated that *B. dorsalis* males that consume ME incorporate metabolites of this compound into their pheromone [11,12], produce a sex pheromone that is highly attractive to conspecific females [13,14], and gain a mating advantage over males deprived of access to ME [15,16]. Similar results have been reported for males of several other ME-responsive, congeneric species, notably the guava fruit fly, *B. correcta* (Bezzi) [15], the carambola fruit fly, *B. carambolae* Drew & Hancock [17,18], and the breadfruit fruit fly, *B. umbrosa* (Fabricius) [19].

Recognition of the interaction between ME and male mating success has prompted research, not only on basic biology, e.g., [20,21], but on applied aspects as well. All of the *Bactrocera* species mentioned above are serious, invasive pests of commercially important vegetables and fruits worldwide [22], and their detection triggers area-wide control and eradication programs [23,24]. For over 60 years, ME has been used in the male annihilation technique (MAT), where ME mixed with insecticide is distributed widely in the environment to eliminate males and, thus, curtail population growth [25]. The link between ME consumption and male mating success further suggests an avenue for improving another important control tactic, the sterile insect technique (SIT). The SIT involves the mass production, sterilization, and release of males of the target species in order to obtain sterile male x wild (fertile) female mating, which yields inviable eggs and subsequently reduces the growth of the wild population [26]. The effectiveness of the SIT depends largely on the mating competitiveness of released, sterile males, and various studies [15,18,27] indicate that the pre-release ME feeding of sterile *B. dorsalis* males boosts their mating competitiveness.

*Bactrocera zonata* (Saunders), which occurs throughout South Asia, the Arabian Peninsula, and northern Africa, is considered a quarantine pest whose hosts include mango, peach, guava, and citrus, among others [28]. The control of *B. zonata* relies primarily on chemical measures, namely, insecticidal cover sprays as well as bait sprays [29,30,31], and, to a lesser extent, the MAT [32,33]. It appears that SIT could be an effective management tool for *B. zonata* [34,35], and artificial diets for mass-rearing are being developed [36]. If the SIT is implemented, pre-release feeding on ME by sterile males may increase their mating ability and thus increase the SIT’s efficacy. *Bactrocera zonata* males are attracted to, and feed on, ME [22], and Rasool et al. [37] recently showed that, as in *B. dorsalis*, ME feeding conferred a mating advantage over males deprived access to the chemical, strongly suggesting that pre-release ME feeding might be a useful procedure.

The objective of the present study was to determine whether feeding on β-caryophyllene (BCP), another widely occurring plant compound [38], likewise increases male mating performance in *B. zonata*. We focused on this alternative compound because ME, despite its potential utility in the SIT, may pose health risks to workers and, thus, be unacceptable for large-scale use. ME has been classified as a carcinogen by The National Toxicology Program, US Department of Health and Human Services, based on research documenting the induction of liver and stomach tumors following the ingestion of ME by experimental animals [39,40]. Here, we describe a series of mating tests intended to (i) confirm earlier results obtained with ME, (ii) assess whether BCP similarly enhances male mating success over untreated males, (iii) compare the copulation frequency between ME- and BCP-fed males in direct competition, and (iv) determine the relative mating success among ME-fed, BCP-fed, and untreated males all competing simultaneously. In addition, because modeling indicates that the concurrent application of the MAT and SIT is more effective in reducing fruit fly populations than their sequential application (i.e., the standard approach; [41]), we determined whether prior feeding on ME or BCP lowered male attraction to ME-baited traps, a requisite for the simultaneous use of the MAT and SIT.

## 2. Materials and Methods

### 2.1. Study Insects

The *B. zonata* flies used in the experiments originated from a colony initiated from infested guava fruits (*Psidium guajava* L.) (Myrtaceae) and maintained at the Insect Pest Management Program, National Agricultural Research Centre, Islamabad, Pakistan (33.67° N, 73.13° E; elevation 515 m). The colony-holding room was maintained at 25 ± 2 °C and 60 ± 5% RH with a photoperiod of L14:D10. At the time of the experiments, the flies were ~40 generations removed from the wild. The rearing procedures followed those described by Rasool et al. [37]. Importantly, for this study, the adults were separated by sex within 24 h of emergence (mating activity commences at 8 d of age [37]) and were provided hydrolyzed yeast mixed with sugar (3:1 ratio by weight) ad libitum. The test flies were held in rooms isolated from the colony-holding room under identical conditions.

### 2.2. ME Feeding

ME (technical grade, 99% purity) feeding was carried out in a room isolated from all other test flies following the methods given by Rasool et al. [37]. Briefly, sexually mature males (15 d old; n = 120) were marked on the thorax with a dot of paint one day and transferred to a plexiglass cage. The next day, 0.5 mL of ME was applied to a piece of filter paper held in a Petri dish and placed in the plexiglass cage at 09:00 h. The marked males (termed ME-fed males) were free to feed on the ME for one hour, after which the Petri dish was removed, and food and water were supplied to the males.

### 2.3. BCP Feeding

The protocol described above for the ME feeding procedure was utilized for BCP feeding, except that only 0.1 mL of BCP was provided.

### 2.4. No-ME and No-BCP Treatment

Untreated males, i.e., those not given ME or BCP, were held in a room isolated from ME-fed and BCP-fed males. The untreated males were marked using the same procedure as the ME- and BCP-fed males and were also supplied with the standard diet and water ad libitum. Untreated males, as well as females, were 16 d old when used in the mating tests.

### 2.5. Field Cages

Circular walk-in field cages (2 m height, 1.5 m diameter) covered with screen nylon mesh (360 μm) were used for the experiments. Four field cages were placed inside each of two adjacent glass houses of the same size (3.96 × 3.96 × 3.81 m), allowing eight replicates per day. A potted artificial citrus tree (1.8 m height, 1 m canopy diameter) was placed in each field cage. Exhaust fans provided ventilation in the glass houses; air temperature was 26 ± 2 °C and humidity was 60 ± 5% RH during testing. Illumination was natural, and no artificial lighting was supplied.

### 2.6. Effect of ME and BCP Feeding on Male Mating Competitiveness

#### 2.6.1. Experiment 1

Males fed on ME competed against untreated males for copulation. Treated males were evaluated for mating competitiveness one-day post-treatment (DPT, i.e., at 16 d of age). Twenty treated and 20 untreated males were released in each field cage approximately 90 min before sunset, and virgin females (n = 20 per cage) were introduced 15 min later. The light levels at twilight ranged between 1400 and 1800 Lux. Upon detection, mating pairs were collected in plastic vials (75 × 30 mm) with screened lids and one person observed two cages. Following the trials on a given day, the mated pairs were brought to the laboratory, where the males were identified. Eight replications were performed.

#### 2.6.2. Experiment 2

The same protocol was followed as in Experiment 1, except that BCP-fed males competed against untreated males for mating. The experiment was replicated eight times.

#### 2.6.3. Experiment 3

Similar experiment protocols were used as in Experiment 1, except that ME- and BCP-fed males competed for copulations in the absence of untreated males. The experiment was repeated eight times. 

#### 2.6.4. Experiment 4

Similar experiment protocols were adopted as in Experiment 1, except that all three male treatments—ME-fed, BCP-fed, and untreated males—competed for mating. As above, 20 males of each treatment and 20 females were released in each cage. Eight replications were conducted.

### 2.7. Effect of ME and BCP Feeding on Male Response to ME-Baited Traps

*Bactrocera zonata* males treated with ME or BCP at 15 d of age were tested for their responsiveness to traps baited with ME at 1, 3, 5, and 7 DPT. Same-aged, untreated males were also tested. The experimental protocol followed Rasool et al. [37]. In short, tests were performed in four nylon mesh cages (each cage L7.9 m × W2.28 m × H1.82 m; mesh size 360 μm) placed inside a larger screen house (L15.24 m × W9.14 m) composed of stainless-steel mesh (390 μm).

One Steiner trap [42], baited with a cotton wick containing 0.5 mL of ME, was placed in each of the four cages. The trap was located at one end of the cage at a height of 1.8 m, and the males (n = 20 for BCP-fed, ME-fed, and untreated, respectively) were released at the opposite end at 08:30 h. Male captures were recorded 3 h later. Temperatures ranged from 28–32 °C during experiments. Eight replicates were conducted.

### 2.8. Data Analysis

Mating success was analyzed using the t-test in Experiments 1 and 2 as the parametric assumptions were met, and the Mann–Whitney test in Experiment 3 as the data were not normally distributed. Experiment 4 involved mating competition among three groups, and a 1-way ANOVA was used to test for significant variation in mating success followed by the Tukey (HSD) test to identify pairwise differences (α = 0.05). The final experiment measured male response to ME-baited traps, and 1-way ANOVA was performed for each test interval (i.e., each DPT) to determine whether ME- or BCP feeding reduced trap captures relative to control males (parametric assumptions were met in all cases). The Tukey test was again used to identify pairwise differences (α = 0.05).

## 3. Results

### 3.1. Mating Competitiveness

The results of the mating experiments are presented in Table 1. Feeding on either ME or BCP conferred a mating advantage over untreated (control) males. ME-fed males obtained 63% of the total matings in Experiment 1, and BCP-fed males achieved 57% of all mating in Experiment 2. No significant difference in mating success was observed in competition between ME- and BCP-fed males and each treatment accounted for approximately 50% of the total matings in (Experiment 3). In the final mating test, ME-fed, BCP-fed, and untreated males competed for females, and significant variation in mating success was detected. Post hoc multiple comparisons revealed that the BCP-fed males obtained significantly more matings than untreated males, while no significant difference was observed between ME-fed and BCP-fed males or ME-fed and untreated males.

### 3.2. Male Response to ME-Baited Traps

There was significant variation in captures among the treatment groups at each of the test intervals (DPT 1: F = 125.2; DPT 3: F = 13.1; DPT 5: F = 17.1; DPT 7: F = 33.9; *p* < 0.001 in all tests; df = 2, 21 in all tests). The Tukey test revealed that ME-fed males were captured in significantly lower numbers than control males at all test intervals, while BCP-fed males were captured less frequently than control males at DPT 1 and 7, but similar numbers to the control males at DPT 3 and 5 (Figure 1). Also, captures of ME-fed males were significantly lower than captures of BCP-fed males at each test interval.

## 4. Discussion

The present study demonstrates that, in trials conducted in semi-natural field cage conditions, the males of *B. zonata* that fed on BCP had a mating advantage over males not given BCP. As noted above, studies [15,17,18,19] on several different *Bactrocera* species have documented a similar effect for males that have fed on ME. However, to our knowledge, only Wee et al. [43], working with *B. correcta*, examined the impact of BCP feeding on male mating success. As in the present study, those authors found that BCP feeding boosted male mating ability and did so to the same extent as feeding on ME. In both our study and that of Wee et al. [43], mating tests were conducted exclusively at 1 DPT, and future tests should investigate how long the positive impact of BCP feeding extends. In *B. dorsalis*, for example, the consumption of ME confers a mating advantage for as long as 35 days after feeding [13].

Why BCP feeding conferred a mating advantage is unknown. In *B. dorsalis* males, ingested ME is transformed into (E)-conferyl alcohol, 2-allyl-4,5-dimethoxyphenol, and trace amounts of (Z)-3,4-dimethoxycinnamyl [11,44]. These compounds are sequestered in the rectal gland and then incorporated into the male sex pheromone [12]. The pheromone of ME-fed *B. dorsalis* males is more attractive than that of the control (unfed) males [13], presumably owing to the presence of the ME metabolites. Data are not available regarding the fate of ingested ME or BCP in male *B. zonata*. However, laboratory studies of *B. correcta* show that ingested ME is transformed into (Z)-conferyl alcohol and (Z)-3,4-dimethoxycinnamyl alcohol, which accumulate in the male rectal gland [45]. Moreover, a large proportion of wild-caught males of *B. correcta* had large quantities of plant-borne, sesquiterpene hydrocarbons, including BCP, in the rectal gland [45]. Thus, it appears that *B. zonata* males (like *B. correcta* males) ingest ME and BCP from plant sources and sequester ME metabolites and/or unmodified BCP in the rectal gland and incorporate them into the sex pheromone.

Trapping studies of *B. correcta* further hint at the role of BCP in the mating biology of *B. zonata* males. Both Tokushima et al. [45] and Wee et al. [43] deployed traps baited with BCP (along with another sesquiterpene, α-humulene, in the former study) or ME and found that both trap types attracted and captured *B. correcta* males. In contrast, *B. dorsalis* males, which also occurred in the trapping area, were found in ME-baited traps exclusively [45]. Although laboratory studies are needed, this finding suggests that *B. zonata* males are more responsive to BCP than *B. dorsalis* males and, accordingly, that BCP may play a more important role in sexual communication in *B. zonata* than in *B. dorsalis*.

Independent of comparisons with the *B. dorsalis*–ME association, the finding that BCP feeding enhanced the mating success of *B. zonata* males indicates that BCP might be used as a pre-release treatment to improve the effectiveness of SIT programs, thus circumventing the potential health risks associated with ME [39,40]. While this is an important first step, subsequent research should investigate whether exposure to BCP volatiles, in lieu of actual ingestion, similarly boosts male mating ability. As outlined previously [16,18], feeding millions of sterile adult *Bactrocera* males performance-enhancing compounds prior to their release is logistically impractical. Tan and Tan [46] designed an apparatus to feed ME to sterile males, where males are brushed off and collected after feeding on an ME-impregnated conveyor belt. While performing well in a research setting, this system has not been tested on an ‘industrial’ scale, where large numbers of males would require treatment on a daily basis. More promising is the approach adopted for the Mediterranean fruit fly, *Ceratitis capitata* (Wiedemann), where exposure to the aroma of ginger root oil is used to increase the mating and dispersal ability of sterile males [47]. Recently, we demonstrated that aromatic exposure to ME, so-called aromatherapy, increased the mating competitiveness of *B. zonata* males [48] (see also Haq et al. [16,18] for similar results with *B. carambolae* and *B. dorsalis*, respectively). As noted, however, unless health risks can be minimized, large-scale ME aromatherapy may not be possible with any *Bactrocera* species. Tests on BCP aromatherapy are planned and, if successful, indicate that BCP could be a viable alternative to ME in developing and improving the SIT against *B. zonata*.

Finally, the result that feeding on BCP reduced subsequent attraction to ME-baited traps suggests that *B. zonata* is a candidate for the simultaneous application of the MAT and SIT. Males fed ME showed significantly lower responsiveness to ME-baited traps than males fed BCP, but BCP-fed males were captured in lower numbers than untreated males, though the difference was not significant at any of the PDTs tested. The higher capture of BCP-treated released, sterile males may represent a necessary cost of reducing the health risks associated with using ME as a pre-release treatment.

In conclusion, the development of effective SIT and MAT programs against *B. zonata* is imperative given the likelihood of increasing restrictions on insecticide use and the aggressive invasiveness of this species. In Africa, for example, climate and host availability models show the significant range expansion of *B. zonata* in both western and southern regions of the continent [49]. Thus, the economic damage caused by *B. zonata* is likely to increase greatly in the coming decades. The present paper explored the possibility that BCP, a safe alternative to ME, might be used to both improve the mating competitiveness of released males in the SIT and allow simultaneous implementation of the SIT and MAT. Based on the promising results obtained, future tests are planned to examine, among other topics, the duration of the BCP-mediated mating advantage, the effectiveness of BCP aromatherapy in enhancing mating ability, and the potential mortality effects of BCP exposure.

## Figures and Tables

**Figure 1 insects-15-00310-f001:**
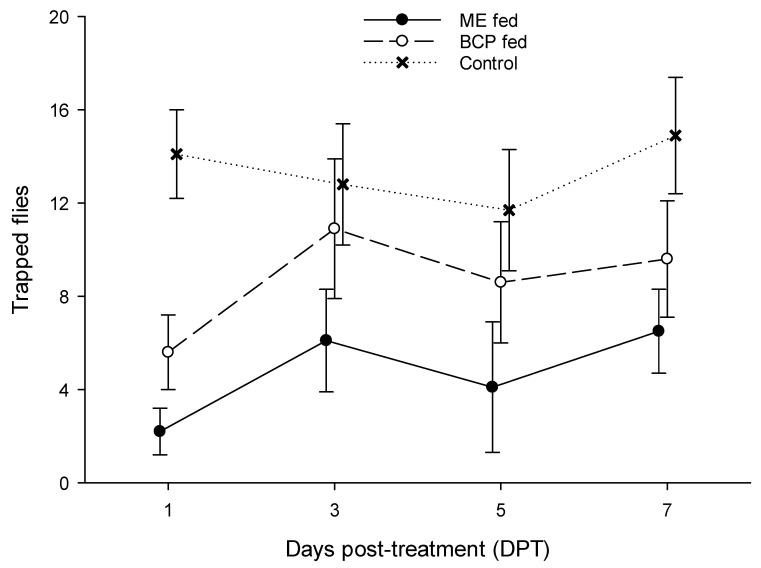
Captures of *Bactrocera zonata* males in ME-baited traps at various intervals after feeding on ME or BCP compared to control (untreated) males. Symbols represent means (±SD) with N = 8 for all male groups.

**Table 1 insects-15-00310-t001:** Results of mating competitiveness experiments. Mating values represent means (±SD) with N = 8 in all cases. The *t*-test was used for Experiments 1 and 2 (df = 14 in both cases), while the Mann–Whitney test (T statistic) was used for Experiment 3 as data were non-normal. Experiment 4 was analyzed using 1-way ANOVA (df = 2, 21).

Experiment	Competing Males	Matings	Test Statistic	*p*
1	ME-fed	11.9 (1.1)	t = 7.31	<0.001
	Untreated	7.0 (1.5)		
2	BCP-fed	9.9 (2.2)	t = 2.44	0.03
	Untreated	7.5 (1.7)		
3	ME-fed	8.9 (1.9)	T = 68.0	1.0
	BCP-fed	9.0 (1.4)		
4	ME-fed	6.0 (2.4)	F = 4.65	0.02
	BCP-fed	8.6 (3.0)		
	Untreated	5.1 (1.5)		

## Data Availability

The original contributions presented in this study are included in the article; further inquiries can be directed to the corresponding author.

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
