# Peer review of "Consumption of β-Caryophyllene Increases the Mating Success of Bactrocera zonata Males (Diptera: Tephritidae)"

_insects, 2024, doi:10.3390/insects15050310_

Round 1

Reviewer 1 Report

Comments and Suggestions for Authors

From my point of view, the introduction should improve its writing and organization, starting from the general to the particular. I suggest putting the objective at the end.

My recommendation is accepted after minor revision

Author Response

Thank you for reading the manuscript and for your encouraging comments. Our response is as follows:

Problem statement: We believe we followed the outline you suggest. That is, we began broadly, with mention of (i) the attraction of male Bactrocera to certain plant compounds, (ii) the impact of ingestion of these compounds on male mating success, and (iii) use of this interaction in control methods. After this, we focused on the possible use of an alternative to methyl eugenol. The first sentence of the final paragraph of the Introduction does not pose a question, but we believe states the study’s objective clearly. It reads: The objective of the present study was to determine whether feeding on β-caryophyllene (BCP), another widely occurring plant compound [38], likewise increases male mating performance in B. zonata.

Methodology: The study was conducted on a small research scale. Potential implementation of BCP feeding/aromatherapy will require – as you rightfully note - much more research on mode of presentation, dose offered, and long-term impacts of BCP on Bactrocera males. We felt it premature to broach these large-scale issues given that our study was a 1st examination of the impact of BCP on male mating success. Much more work awaits!

Discussion: As above, our results derive from the first ever examination of the BCP-B. zonata interaction, which was a small-scale research project. Use of BCP in control programs requires considerably more research. As you suggested, we noted the need to measure the duration of any BCP effect on male mating success (Line 238 of original paper) and any mortality effects (Line 300 of the original paper).

Reviewer 2 Report

Comments and Suggestions for Authors

A very well written and bibliographically supported paper is presented on the potential use of a plant product chemical compound on Tephritid SIT and/or MAT programs. The results strongly support the originality of the authors' idea and English language is of high level.

I recommend the publication of the paper and only a few errors/comments are stated in the pdf file.

Author Response

Thank you for reading the manuscript and your comments.

Regarding the Tukey test, we used what is typically referred to as the Honestly Significant Difference (or HSD) test. We have inserted (HSD) when the Tukey test is first mentioned.

Regarding Figure 1, the legend noted that the error bars represent SD.

Reviewer 3 Report

Comments and Suggestions for Authors

The manuscript “Consumption of β-caryophyllene increases the mating success of Bactrocera zonata males (Diptera: Tephritidae)” reports semi-field trials on the evaluation of mating performance of males fed Methyl eugenol or β-caryophyllene. Trapping efficacy tests were also carried out on traps baited with Methyl eugenol using males fed with Methyl eugenol or β-caryophyllene. While there are several studies in this field using Methyl eugenol, little is known about β-caryophyllene. Moreover, the authors justify this research given the health risks of manipulate methyl eugenol.

I think research in this field is needed. The introduction of the paper is concise, the method applied is appropriated and the results are clear. The discussion are good.

Here are some suggestions:

Lines 5-6: authors are reported two times.

Line 15: define ME.

Line 15: ...Bactrocera males. spp... change with: ....Bactrocera spp. males...

Line 58: delete “(“

Line 76: for reference 37 add the authors. Rasool et al. (2023) [37] recently....

Lines 135-139: the dimensions of the circular cages and glass houses are reported twice. One line is missing. Please rephrase these sentences.

Section 2.6: in experiments 1-4, how long do the specimens stay in the cage? Please add this in the text.

Line 196: the term “Experiment 5” is mentioned here for the first time. Please cite it by this name in section 2.7

Table 1: for easier interpretation, please add in the Experiment 4 the result of the Tukey test.

Figure 1: the line graph would suggest flight curves with captures over time. In this case, a graph with histograms or a table would be more appropriate. As with Table 1, the results of the Tukey tests should also be added.

Author Response

Thank you for reading the paper and for your comments. Our response is as follows:

Lines 5-6 – duplication of authors’ names seems to be an error associated with the journal’s formatting and layout process

Line 15 - the acronym ME has been inserted when methyl eugenol is first mentioned, thus clearing up the problem you noted

Line 15 – changed to Bactrocera spp. males as suggested

Line 58 – parenthesis deleted

Line 76 – Rasool et al. [37] inserted

Lines 135-139 – duplicate dimensions were removed; wording improved to increase clarity

Section 2.6 – observations were made until 15 min after sunset. This information has been added.

Line 196 – The ‘Experiment 5’ label is now explicitly mentioned in the Methods section

Table 1 – results of the Tukey test were inserted in the table using superscript letters to designate statistical differences

Figure 1 – we retained the original presentation but results of the Tukey test were added using letters to designate statistical differences

Reviewer 4 Report

Comments and Suggestions for Authors

Manuscript ID: insects-2969781entitled “Consumption of β-caryophyllene increases the mating success of Bactrocera zonata males (Diptera: Tephritidae)” by Ihsan Haq, Sehar Fatima, Awais Rasool, and Todd Shelly submitted to Insect Pest and Vector Management is a well written manuscript from start to end. I recommend consideration for publication after minor revision. I made some edits in the attached PDF for authors’ revision, some of the notes should be discussed are:

·         β-caryophyllene should be characterized further by its source.

·         β-caryophyllene is not safe as stated in the manuscript. This chemical causes irritation of the skin. When heated this compound emits acrid smoke and irritating fumes.  It is considered a compound with toxicity at doses higher than 2000 mg/kg body weight.   Safety: chrome-extension://efaidnbmnnnibpcajpcglclefindmkaj/https://www.sigmaaldrich.com/US/en/sds/ALDRICH/W225207 .  

Safety: chrome-extension://efaidnbmnnnibpcajpcglclefindmkaj/https://www.agilent.com/cs/library/msds/TRP-115-1_NAEnglish.pdf  

·         The protocol described above for ME feeding procedure was utilized for BCP feeding, except that only 0.1 mL of BCP was provided (line 127). The reason for the different feeding is not mentioned.

·         Marked males in experiments not mentioned from different feeding.

·         Weather conditions (rainy, cloudy, precipitation, wind) and illumination (in lux) in the cages should be mentioned.

Author Response

Line # refer to those in original submission.

Line 4 – this duplication appeared to derive from the journal’s formatting/layout process

Line 15 – corrected

Line 20 – zonata inserted

Line 25 – added ‘when mixed with an insecticide’

Line 52 – replaced Artocarpus with breadfruit

Line 76 – added Rasool et al.

Line 92 – parenthesis added

Line 99 – Coordinates and elevation inserted as suggested

Line 103 – Yes, the diet is the same as for medfly; no change made in text

Line 105 – this sentence was omitted in revised paper

Line 108 – wording changed in revision but reference [37] now cited

Line 115 – this sentence omitted in revised paper

Lines 118, 122 – Petri now used

Line 127 – Based on earlier research (Haq et al. 2018), a dosage of 0.5 mL of methyl eugenol was administered, which proved adequate to satisfy the males without leading to the toxicity that might result from an excessive amount. Regarding BCP, since there is no existing literature on BCP feeding, the decision was made to use 0.1 mL of BCP, which was the amount needed to fully saturate the filter paper. This quantity was found to be effective in improving male mating success, eliminating the necessity to experiment with larger volumes. Nonetheless, future studies will need to determine the appropriate dosage for a feeding protocol in a sterile male release facility. No change was made to the text.

Lines 125, 130 – addressed in Section 2.6.1

Line 135 – mesh size inserted as suggested

Lines 135-139 – duplicate measurements deleted; wording improved.

Line 143 – Light levels at twilight were inserted in section 2.6.1

Line 151 – size of vials inserted as suggested

Line 170 – mesh size now included as suggested

Line 174 – mesh size now included as suggested

Line 175 – mesh size now included as suggested

Line 179 – Light levels were not recorded during this experiment

Line 190 – wording changed as suggested

Line 213 – (+ SD) was retained

Line 214 – N was changed to n

Line 216 – df now added for experiments 1 and 2 in Table 1 legend

Line 227 – N changed to n

Line 306 – affiliation added for S.H. Shah

References – all corrections made